# Brain Plasticity Mechanisms Underlying Motor Control Reorganization: Pilot Longitudinal Study on Post-Stroke Subjects

**DOI:** 10.3390/brainsci11030329

**Published:** 2021-03-05

**Authors:** Marta Gandolla, Lorenzo Niero, Franco Molteni, Elenora Guanziroli, Nick S. Ward, Alessandra Pedrocchi

**Affiliations:** 1NearLab@Lecco, Polo Territoriale di Lecco, Politecnico di Milano, Via Gaetano Previati, 1/c, 23900 Lecco, Italy; lorenzo.niero@mail.polimi.it (L.N.); alessandra.pedrocchi@polimi.it (A.P.); 2Department of Mechanical Engineering, Politecnico di Milano, Via Privata Giuseppe La Masa, 1, 20156 Milano, Italy; 3Villa Beretta Rehabilitation Center, Valduce Hospital, Via N. Sauro, 17, 23845 Costa Masnaga, Italy; fmolteni@valduce.it (F.M.); eleonora.guanziroli@gmail.com (E.G.); 4Department of Movement and Clinical Neuroscience, UCL Queen Square Institute of Neurology, London WC1N 3BG, UK; n.ward@ucl.ac.uk; 5The National Hospital for Neurology and Neurosurgery, Queen Square, London WC1N 3BG, UK; 6NearLab, Department of Electronic Information and Bioengineering, Politecnico di Milano, Via Giuseppe Ponzio, 34/5, 20133 Milano, Italy

**Keywords:** fMRI, carryover effect, functional electrical stimulation (FES), dynamic causal modeling (DCM), parametric empirical bayes (PEB)

## Abstract

Functional Electrical Stimulation (FES) has demonstrated to improve walking ability and to induce the carryover effect, long-lasting persisting improvement. Functional magnetic resonance imaging has been used to investigate effective connectivity differences and longitudinal changes in a group of chronic stroke patients that attended a FES-based rehabilitation program for foot-drop correction, distinguishing between carryover effect responders and non-responders, and in comparison with a healthy control group. Bayesian hierarchical procedures were employed, involving nonlinear models at within-subject level—dynamic causal models—and linear models at between-subjects level. Selected regions of interest were primary sensorimotor cortices (M1, S1), supplementary motor area (SMA), and angular gyrus. Our results suggest the following: (i) The ability to correctly plan the movement and integrate proprioception information might be the features to update the motor control loop, towards the carryover effect, as indicated by the reduced sensitivity to proprioception input to S1 of FES non-responders; (ii) FES-related neural plasticity supports the active inference account for motor control, as indicated by the modulation of SMA and M1 connections to S1 area; (iii) SMA has a dual role of higher order motor processing unit responsible for complex movements, and a superintendence role in suppressing standard motor plans as external conditions changes.

## 1. Introduction

Strokes are one of the main causes of long-term disability worldwide. In fact, as most patients survive the initial injury, the biggest drawback is usually through long-term impairment. When coming to motor impairment, most post-stroke patients recover at least some of their lost motor functions, though the degree of this recovery is variable, depending on several factors including the severity of the damage, the type and intensity of rehabilitation therapy and the commitment of the subject [1,2].

When dealing in particular with Functional Electrical Stimulation (FES) treated foot drop issue, the application of the peroneal nerve stimulation has a positive well-known orthotic effect [3], but a proportion of patients are able to relearn the ability to voluntarily dorsiflex the ankle without the device, as first observed by Liberson and colleagues [4]. This phenomenon, referred to as the “carryover effect”, has been observed in a number of subsequent studies [5,6], even though only a fraction of patients are able to achieve a stable recovery. However, the working mechanism of the carryover effect has not yet been fully understood, although it has been hypothesized to have its roots in a brain plasticity mechanism. In fact, FES delivered to a nerve tract containing both efferent motor and afferent sensory fibers synchronously depolarizes motor and sensory axons that are bundled together, eliciting muscle contraction through two pathways: direct motoneuron depolarization, and indirect mechanism, providing excitatory synaptic input to spinal neurons that recruit motor units in the natural order [7]. Therefore, FES experimentally actuates reafference (i.e., the sensory input that results from the movement).

Brain mapping studies in post-stroke patients have revealed that the brain reorganizes itself in response to sensory inputs and experience [8,9]. By means of functional Magnetic Resonance Imaging (fMRI) studies, it has been hypothesized that the carryover effect phenomenon has its bases in the interaction between volitional effort and the electrical stimulation proprioceptive information, resulting in a neuroplastic effect on the central nervous system [10,11,12]. By means of fMRI experiments with a 2 × 2 factorial design, with volitional effort and FES as experimental factors, it has been hypothesized that the mechanism through which FES carryover takes place in post-stroke subjects is based on movement prediction together with a sense of body ownership [13]. In other words, in patients showing FES carryover (responders), the execution of the movement with concurrent volitional intention and FES allows to correctly plan the movement and to perceive it as self-generated. The execution of a voluntary movement, in fact, requires the brain to integrate both the volitional intention to execute the movement and the knowledge about the state of the body (i.e., integrate sensory feedback). By doing so, the motor control loop correctly updates itself [14], showing a long-lasting formation or strengthening of new functional connections following Hebbian principles, through the combination of volitional effort and the sensorial perception of a properly completed movement [15,16]. In humans, changing proprioceptive input influences motor cortex excitability [17,18]. Conversely, the response of somatosensory cortex neurons to proprioception is modified by the motor task [19]. 

This work aims at understanding the underlying connectivity network of the carryover effect, by longitudinally (pre and post treatment) evaluating differences between groups of patients with and without FES carryover (i.e., responders and non-responders), and in comparison with a healthy control group. From the analysis of parameters estimates of Dynamic Causal Modeling (DCM), we aim at discriminating the behavior of patients showing or not the carryover effect and to highlight any longitudinal changes in effective connectivity elicited by the rehabilitation program.

## 2. Materials and Methods

### 2.1. Participants

Patients were recruited from the Villa Beretta Rehabilitation Centre (Costa Masnaga, LC, Italy). All patients had suffered from first-ever stroke >6 months previously, resulting in weakness of at least the tibialis anterior muscle (up to 4 + on the Medical Research Council scale). Exclusion criteria consisted of: (i) responsiveness of less than 10° in FES-induced ankle dorsiflexion; (ii) language or cognitive deficits sufficient to impair cooperation in the study; (iii) inability to walk even if assisted; (iv) high spasticity at ankle joint plantar flexor as measured by the modified Ashworth scale index >2. The age-matched control group was composed of healthy volunteers with no neurological or orthopedic impairment. The procedure has been approved by the Villa Beretta Rehabilitation Centre Ethics Committee, and all subjects gave informed written consent.

### 2.2. Experimental Set-up

The experimental set-up included a 1.5 T MRI scanner (GE Healthcare, Cv/I, Chicago, IL, USA), a motion capture system (Smartμg; BTS Bioengineering, Garbagnate Milanese, Italy) and an electrical stimulator (RehaStim pro; Hasomed GmbH, Magdeburg, Germany), as previously described and validated [20,21].

### 2.3. Patients Functional Assessment and Foot Drop Rehabilitation Procedures

After enrollment, patients underwent a baseline assessment protocol, which included gait velocity and paretic step length, endurance velocity (6 min walking test), the Tibialis Anterior activation index derived from electromyographic measurements [22], and Medical Research Council index at ankle joint. All patients underwent the same rehabilitation treatment, based on FES of the Tibialis Anterior muscle for the recovery of the drop foot, lasting 4 weeks [6]. At the end of the treatment, a second assessment session was performed, with the baseline outcome measures. Patients were classified as responders or non-responders, based on whether or not they achieved the carry-over effect, by combining the outcome measures with a dedicated algorithm successfully validated against clinicians’ assessment [6].

### 2.4. fMRI Paradigm

The same fMRI assessment session has been performed once by controls and twice by patients, i.e., before and after the rehabilitation program. A 2 × 2 event-related fMRI design, with volitional intention (with the levels volitional and passive) and FES (with stimulator on or off) factors was performed using right ankle dorsiflexion. During a continuous 10 min scanning session, subjects performed 20 alternating 9 s OFF and 21 s ON blocks with the following conditions: (i) FES-induced ankle dorsiflexion concurrently with voluntary movement; (ii) FES-induced ankle dorsiflexion, while the subject remains relaxed; (iii) voluntary ankle dorsiflexion; (iv) passive dorsiflexion (by the experimenter). The subjects were instructed to remain relaxed when no volitional contribution was required, and to equally voluntarily contribute when requested. Movements were paced every 3.5 s with an auditory cue. All subjects were free to choose the amplitude of their active movement to preclude fatigue. The experimenter moved the ankle to match in terms of amplitude the movements during volitional dorsiflexion. Subjects were instructed to keep eyes closed and head movements were minimized with rubber pads and straps; knees were bent with the subject’s legs lying on a pillow. Functional electrical stimulation was applied to the peroneal nerve through superficial self-adhesive electrodes, with biphasic balanced current pulses at 20 Hz fixed frequency. The pulse width had a trapezoidal profile (maximum pulse width 400 µs) and the current amplitude was set subject by subject so as to reproduce the same movement amplitudes as during voluntary movements, within the tolerance threshold. Current amplitude and pulse width were kept the same for both FES-based conditions.

### 2.5. fMRI Image Acquisition and Preprocessing

T1-weigthed anatomical images (0.94 × 0.94 × 0.94 mm voxels) and T2*-weighted MRI transverse echo-planar images (1.8 × 1.8 × 4 mm voxels, TE = 50 ms) were acquired, with repetition time equals to 3 s. Each echo-planar image comprised 22 contiguous axial slices, positioned to cover the temporo-parietal and occipital lobes. Due to technical reasons, it was not possible to acquire the cerebellum. The first six volumes were discarded to allow for T1 equilibration effects. A total of 200 brain volumes were acquired in a single run lasting 10 min. Activation patterns for healthy and patients’ groups have been reported previously [13], and summarized in the Results section.

### 2.6. Regions of Interest (ROIs) Selection and Extraction

The functional images were smoothed with an isotropic 4 mm full-width half-maximum kernel for dynamic causal modeling [23] with SPM12 software. The general linear model was reformulated to specify the driving and modulatory experimental inputs:driving input representing descending voluntary signals—V (combined onsets of the two conditions requiring volitional effort);modulatory input encoding the contribution of ascending functional electrical stimulation to proprioceptive input—E (combined onsets of the two conditions requiring stimulation);driving input representing underlying proprioceptive input from all movements—P (onsets from all conditions). 

The selection of the ROIs was made on the basis of prior knowledge about the role of cortical areas in motor control loop.


Leg section of contralateral primary sensorimotor cortices (M1 and S1), known to have a fundamental role in the motor control loop, especially in the sensorimotor integration mechanism [24,25].Supplementary motor area (SMA), known to have a role in the processing of voluntary action; it has long been thought to play a special role in the internal generation of complex movements and to be responsible for higher order aspects of motor planning [26,27].Angular gyrus (AG). AG is assumed to be a multi-sensorial integration hub [28], also able to process discrepancies between intended action and movement consequences, in such a way that they can be consciously detected by the subject [29]. Moreover, in an earlier study [13], AG and SMA have been shown to be differently active in responders and non-responders.


The selection of subject-specific ROI coordinates was informed by the SPM12 atlas of maximum probability tissue labels (Neuromorphometrics.com). For each region, subject-specific maxima were selected within the activation map, after the corresponding mask was applied. SPM12 atlas contains masks labeled for SMA and AG, but specific masks for M1 and S1 “leg area” were not present. In these cases (M1 and S1), statistical parametric maps were masked using “pre-central gyrus” and “post-central gyrus” labeled masks, and activation maxima were manually selected within these masked maps. For each subject, regional responses were summarized with the first eigenvariate of a 4 mm radius sphere centered in the subject-specific maxima. The selected maxima, from the 4 different ROIs have been checked to have a Euclidean distance of at least 8 mm from one another.

### 2.7. Dynamic Causal Modeling (DCM) Analysis

The observed fMRI time series are used to estimate directed (effective) connectivity among regions or nodes [23], following dynamic causal modeling equation to describe the evolution of neuronal activity ($\dot{x}$) in terms of a mixture of inputs from other areas and experimental inputs (*u*) as follows:(1)$$\dot{x}=\left(A+\sum u_{j}B^{j}\right)x+Cu$$

The dynamic causal modeling includes the estimation of the following matrices.


matrix A—average connectivity among brain regions during the experiment, irrespective of task modulation (endogenous connections).matrix B—change in endogenous connections that can be elicited by an experimental variable (modulatory inputs).matrix C—direct influences of an experimental variable on specific regions (driving inputs).


Model inversion returns an approximation to model evidence, obtained by the free energy approximation. This can then be used in Bayesian model selection [30] to identify the model that is most likely, given the observed data, among a set of competing models.

### 2.8. Parametric Empirical Bayes (PEB) Model

The Parametric Empirical Bayes (PEB) framework involves the creation of a hierarchical model consisting of a first level modeling within-subject effects (subject specific DCM), and a second level modeling between-subject or group effects [31,32]. At the second level, a general linear model is used to model first level coupling parameters as random variables, oscillating around a group mean with additive Gaussian noise.

As for the analysis of model structure in healthy controls, we assumed there were no subgroups, and thus the general linear model at the second level was modeled using a single mean regressor. On the other side, patients were regressed as responders (i.e., showing carry-over, CE subgroup) and non-responders (nCE), with two fMRI scans each—one before (PRE), and one after (POST) the rehabilitation treatment. Therefore, besides a group mean regressor, a carry-over regressor, encoding the carryover classification (CE = 1, nCE = −1) and a time regressor, encoding the acquisition time point (PRE = 1, POST = −1) have been included. Moreover, the capacity score regressor (i.e., comprehensive score to assess residual ability which combines all acquired outcome measures [6] has been included to take into account for the inter-subject variability associated to the residual motor capability. Subject-specific capacity scores were mean-corrected and normalized between [−1, 1], before being included in the design matrix.

### 2.9. Model Structure Identification Procedure

Given that the high number of parameters (connections) to be tested would give rise to such a vast model space that it would be difficult to even define all possible competing models [28,33,34], a hierarchical approach has been defined. Similar to the approach described by Friston and colleagues [32], inference on model space was carried out one matrix at a time, through Empirical Bayesian model reduction, followed by model comparison and eventually performing a Bayesian Model Average over the best performing models (Figure 1).

STEP 1: input matrix definition for controls group. The C matrix model space was based upon prior knowledge about functional anatomy. Input V was assumed to drive either M1 (Model 1), SMA (Model 2) or both (Model 3), modeling top-down intentional signals during voluntary movements, input P was assumed to drive S1, modeling the proprioceptive and somatosensory consequences of movement (e.g., ascending afferents from muscle spindles and Golgi tendon), known to convey information to sensory areas. Winning model was determined through Bayesian Model Selection by comparing model evidences. C matrix was then fixed as equal to the winning model of this STEP for all subsequent analyses.

STEP 2: structural matrix definition for controls group. To address the problem of high number of models in the model space, we introduced constraints, based on prior knowledge about anatomical and functional connections. Some connections have been considered as fixed: (i) M1 and S1 bidirectional connections, by virtue of their anatomical connections [35,36]; (ii) SMA to M1 anatomical projections have been reported in literature [37,38]; (iii) M1 to SMA connection was assumed to be a functional feedback connection, through basal-ganglia-thalamo-cortical circuits, which are known to be crucial for the initiation and control of voluntary action [37]. All the other bidirectional connections were tested. These models only differed in terms of matrix A, as C matrix has been derived from STEP 1, and B matrix was assumed as fully connected. The optimal model was obtained applying a BMA over the best performing models [39].

STEP 3: bilinear matrix definition for controls group. Input E (i.e., electrical stimulation) was the only input allowed to modulate intrinsic connections. An exhaustive search was performed, producing a Bayesian model average as output [32].

STEP 4: input matrix definition for patients. C matrix structure was assumed fixed as the result from STEP 1, except having possibly different weights over the different existing connections.

STEP 5: structural matrix definition for patients’ group. It was allowed to discard one or more intrinsic connections, starting from the structure obtained for healthy subjects (i.e., it has been hypothesized that patients cannot exhibit effective connections that are missing in the controls’ group). An exhaustive search at second level over the entire model space was applied producing a Bayesian model average as output [32].

STEP 6: bilinear matrix definition for patients’ group. The B matrix structure was investigated starting from the A matrix structure output of STEP 5. Again, an exhaustive search was employed, producing a Bayesian model average [32].

## 3. Results

### 3.1. Participants

The healthy control group was composed by 16 healthy volunteers (8 males), right handed, with no neurological or orthopedic impairment (mean age 36.4 ± 13.8 years). Their results in terms of activation maps have been fully reported previously [12]. Overall, 14 post-stroke patients (8 males) were recruited (mean age 44 ± 14) (Table 1), whose pre-treatment activation maps analysis has been previously reported [13]. In this study, 8 patients who presented 2 fMRI sessions, i.e., one before (PRE) and one after (POST) the rehabilitation treatment, have been included (Table 2).

### 3.2. Activation Patterns for Healthy and Patients’ Groups (Previous Results)

The analysis of the activation pattern obtained by the healthy controls and the patients groups has been previously reported [13]. However, for the sake of clarity of the results obtained by the connectivity analysis, the main findings are hereby summarized.

(i) Healthy controls show clear activation in all conditions in motor and somatosensory areas known to be involved in ankle dorsiflexion execution, as expected.

(ii) Patients and controls primarily show common activation in bilateral sensorimotor (all conditions) and supplementary motor (for conditions where volitional intention is present) areas. Compared to the control group, patients tend to over activate right AG when there is no volitional intention to move and left intraparietal sulcus during passive movement.

(iii) We observed a quite consistent separation between FES responders and non-responders on the basis of the brain responses in SMA, M1, and AG regions acquired before the treatment, where patients with FES carryover appear to have “normal” responses, whilst those without do not. In particular, patients who experienced the carryover effect have responses in SMA and M1 that correspond to healthy controls, whilst responses in these regions in patients without carryover are diminished. Conversely, the interaction between factors in contralateral AG is seen only in those without carryover but not in those with carryover or healthy controls.

(iv) We suggest that the mechanism of action of FES carryover is based on movement prediction (mediated by SMA area) and sense of agency or body ownership (mediated by AG area). In other words, the ability of a patient to plan the movement and to perceive the stimulation as a part of his or her own control loop is important for the FES carryover effect to take place.

### 3.3. DCM Results

The coordinates of ROIs were consistent across subjects, being the standard deviations less than 12 mm in all directions, which correspond to a dispersion of the identified peaks of activation of 3 cubic voxels 3 × 3 × 3 mm (Table 3). Given the PEB model approach, each estimated parameter is composed of a single contribution for the healthy control group (i.e., mean), and by four contributions for the patients group—mean, carryover, time, and capacity score. To obtain the overall magnitude of a connection, it is needed to sum different contributions. As a way of example, for connection from SMA to M1 of patients’ B matrix (Figure 3), for CE group PRE treatment we obtain (+) × −0.39+(+) × 0.51 = 0.12, where the first (+) is encoding the carryover classification (CE = 1, nCE = −1) and the second (+) is encoding the acquisition time point (PRE = 1, POST = −1). Therefore for CE group POST treatment we obtain (+) × −0.39+(−) × 0.51 = −0.9.

As for the control group, the posterior probabilities of Model 3 for the input matrix is 0.81, indicating with strong evidence Model 1 as winning model [40]. Model 1 indicates the presence of input V driving both M1 and SMA activities (Figure 2A). The structural model (i.e., A matrix) with the highest posterior probability (i.e., 0.59) discarded bidirectional connections between AG and M1, and AG over SMA connection (Figure 2C). As for B matrix structure, the exhaustive search indicated all connections but S1 to AG, S1 to SMA, and M1 and S1 self-connections to be modulated by the presence of FES (Figure 2E). When coming to patients, starting from A matrix structure of controls, bidirectional connections between S1 and AG, connection from AG to SMA and from M1 to S1 were discarded, following BMA (Figure 2D). As for patients’ B matrix structure, FES input was observed to modulate all connections, but SMA to AG, S1 to M1 and bidirectional connections between S1 and AG (Figure 2F).

Figure 3 and Figure 4 fully report the estimated parameters for input matrix, intrinsic and extrinsic connectivity matrices for healthy controls and patients.

## 4. Discussion

When dealing with motor rehabilitation of neurological patients, the recovery of a lost motor function is a consequence of a plasticity process, which takes advantage of surviving redundant neural circuits to reprogram neuromuscular activation patterns, in order to perform functional tasks. This is the case of Functional Electrical Stimulation, where the so-called carryover effect, i.e., long-term functional improvement after FES-based therapy, has been observed in sub-groups of patients, and it has not been fully explained by behavioral outcome measures (e.g., degree of initial disability, location of the brain damage, etc.). Implementation of FES in clinical practice would benefit from the understanding of the underpinning mechanisms of action of the carryover effect, possibly identifying biomarkers for a successful outcome of the treatment. This study investigates the longitudinal changes in motor control loop mechanisms, influenced by FES-based treatment for foot-drop correction, and the underlying key features of carry-over effect, by evaluating differences among healthy subjects and groups of patients, distinguishing between responders and non-responders.

The results can be discussed in light of the active inference account for motor control [41] that models the brain as a hierarchical inference engine, trying to optimize probabilistic representations of what caused its sensory inputs. When coming to movement control, top-down predictions are predictions of the sensory consequences of movement. Proprioceptive prediction errors are generated at the level of the spinal cord and result in the activation of motor neurons through classical reflex arcs. Therefore, under active inference account, given the same task and the same state of the system, top-down signals are context-dependent predictions [42]. In our model, M1 and S1 act as low-level areas, representing system input-output nodes. Input is the ascending sensory feedback from the peripheral proprioceptors reaching S1, while output is the descending prediction signal (desired proprioceptive consequences) sent from M1 through the pyramidal tract. SMA and AG are interpreted to be higher order areas [27,43], responsible for higher processing of predictions and prediction errors.

### 4.1. Proprioceptive Input to S1 Area

The proprioceptive input to S1 results to have a lower weight in FES non-responders (Figure 4A), indicating minor ability to exploit and integrate the proprioceptive information in the motor control loop. This is in line with the hypothesis that motor control learning failure is due in some cases to a proprioception deficit [44]. Note that none of the patients showed a higher sensitive deficit before the experiment, leading to the hypothesis that, given the same external input, it is the given weight in the connectivity model which is different between responders and non-responders. In addition, we observed that the more a patient has residual motor capability, the lower is the effect of voluntary input on SMA (Figure 3), going towards values more similar to healthy controls, in line with previous longitudinal and cross-sectional studies on activation changes during motor recovery [45,46].

### 4.2. Influence of SMA and M1 over S1

SMA and M1 are the main responsible for the processing of motor plans with their outgoing connections carrying the proprioceptive prediction of the voluntary movement. With no stimulation in healthy subjects, S1 receives a suppressive influence from M1 (Figure 4E), which we hypothesize to be related to the sensory suppression mechanism, widely reported in literature [47,48]. On the other hand, S1 receives an excitatory influence from SMA (Figure 4F), which might indicates that SMA sends the novel proprioceptive prediction to S1 to raise awareness about new features of the movement consequences. During FES in healthy controls, the activation of S1 promoted by M1 activity (Figure 4E) may be due to FES reversing fibers recruitment order (i.e., large motor units are recruited before small motor units). The superimposition of FES onto voluntary movement can activate more motor units than a voluntary action alone, engendering an increase in the somatosensory consequences of the movement [49]. Moreover, the pairing of voluntary movement and electrical stimulation has been shown to increase cortical excitability over 1 [50]. We observed a concomitant balancing suppressive activity of SMA over S1 (Figure 4E), to prevent overactivation. It can be speculated that SMA informs S1 about the expected effects of FES supported movement, implementing a reprogrammed sensory suppression performed at higher level.

As for the patients’ group, we observed a lack of influence of M1 over S1 (Figure 4E), which might be due to a malfunction in the generation of proprioceptive predictions by M1, given that the patient is not able to perform the required movement, leading to the absence of the consequent sensory suppression (as no sensory consequences are expected). At a higher hierarchical level, SMA stimulates S1 activity (Figure 4F), sending information about residual proprioceptive predictions, trying to enhance S1 excitability as a hub for possible proprioceptive information coming from the periphery. When FES is applied, we observed a negative modulation of the connection from M1 to S1 (Figure 4E), which goes towards the suppressive activity observed in healthy subjects in normal conditions. It can be hypothesized that FES allows to complete the movement (which otherwise they would not be able to perform) and to perceive it as correctly executed, as it would have been for a healthy subject in normal conditions.

### 4.3. Outcoming Connections from S1

With no FES applied, we observed a positive influence of S1 over primary and supplementary motor areas (Figure 4G,H), which may represent the prediction error evaluated and sent back from S1. These connections maintain the same magnitude between CE and nCE groups but, given that the proprioception input has a lower weight in nCE patients, S1 exerts a smaller influence on M1 and SMA. As this would represent the error sent by S1 and used to update predictions in motor areas, a reduced sensitivity to proprioception, resulting in a mistaken predictive error, causes a possible misbehavior of the overall updating mechanism. With FES, in healthy subjects we observed a modulation of S1 to M1 connection, but not of S1 to SMA connection (Figure 4G). In this case, FES produces a movement that might be perceived as different from the usual one, the error computed by S1 is incremented and sent to M1 in order to update proprioceptive predictions. Conversely, the connection from S1 to SMA is not modulated, as the higher-level movement programming remains unchanged, given the easiness of the required task. In patients, the opposite behavior is observed, as only S1 to SMA connection is modulated (Figure 4H). This suggests that only the higher level planning of the movement is influenced by FES, as the subject empowered by FES can complete a task not previously possible, which is perceived as correctly executed as part of his or her control loop. No extra-error is sent back to M1, resulting in the lack of an update of movement consequences prediction at a lower level.

### 4.4. SMA-M1 Bilateral Connections

SMA seems to exert a suppressive activity with respect to M1 in the control group (Figure 4D). We know from literature that SMA exerts a suppressive action over primary motor areas, whenever a perturbation of standard external conditions in favor of alternative predictions planned by SMA is present [37,51]. Given that the experimental context represents an unusual situation, this might be the cause for the observed suppressive activity of SMA in absence of electrical stimulation. In addition, the presence of FES, introducing a further disturbance of standard conditions, enhances this suppressive effect (Figure 4D). In patients, SMA preserves a comparable inhibitory role over M1 for what it concerns A matrix (Figure 4D). When applying FES, in responders SMA preserves its suppressive role (even if of smaller entity with respect to healthy controls), while in non-responders it has an excitatory role. CE patients therefore reflect the behavior of healthy subjects (Figure 3), in agreement with literature [13] about the similarity in activation patterns of SMA between responders and healthy subjects. It can be hypothesized that responders are more keen in considering FES as helper for the completion of movement, and therefore to consequently update the internal predictive motor program. After the treatment, the influence of SMA over M1 becomes inhibitory for both groups, even if it is a larger effect for the CE group. We can suppose that the more the patient practices with FES, the more this suppressive mechanism is enhanced, as they are more capable to consider the disturbance of external conditions in their control loop, or from another point of view, they can take advantage of the support of FES. The higher enhancement observed for CE group (inhibitory effect larger than healthy controls group) might represent a strong motor learning effect suppressing an incorrect motor scheme (e.g., compensatory movements) in favor or the correct one. We observed consistent negative values in both healthy and patient groups for M1 to SMA connection (Figure 4C), both for A and B matrices. The interpretation of this connection is unclear, due to its indirect nature, even if known to be crucial for the initiation and control of voluntary action [37,52]. The estimated values ensure the presence of a negative feedback from M1 to SMA, in agreement with what has been reported by Kasess and colleagues during motor imagery [37].

### 4.5. Role of AG Area

Even if AG has been identified as possible important biomarker area for carryover, the interpretation of AG role is not straightforward, due to the versatility and complexity of its functions and due to the absence of direct structural connections with other regions of the model [28,53]. A major observation may lay on effective connections coming from SMA (proprioceptive prediction) and S1 (prediction error) indicating that AG has, at some extent, a role in calculating the discrepancy between intended action, and relative movement consequences (Figure 3). Similarly, in patients, the estimated parameters associated to the capacity-score regressor indicate that the more the patient has residual ability, the more these connections behave as those of a health subject (Figure 3). We observed that nCE patients present a reinforced positive extrinsic self-connection with respect to CE patients (Figure 3), which can be linked to the increased activation of AG in nCE patients previously reported [13].

### 4.6. Study Limitations

Acquired brain volumes were missing the cerebellum area, which is considered to give a major contribution in the generation of motor commands, and on plasticity due to motor learning and recovery [54]. In addition, images have been acquired on a scanner used in the clinical setting, and therefore not provided with most recent research acquisition protocols. Another important limitation regards the number of subjects involved. Although with a small number of subjects, the presented results are statistically significant, and included in a well-established methodological framework as a possible seed toward a better understanding of the motor-related adaptive plasticity. In future studies, new subjects should be recruited for the further investigation of the presented hypotheses, possibly in a multi-centric setting, and different approaches to induce motor relearning might be investigated, such as the use of exoskeletons [55].

## 5. Conclusions

We presented a DCM-based hierarchical approach including M1, S1, SMA, and AG areas for the identification of brain network structure underlying foot dorsiflexion motor control. Our results point towards the following hypotheses: (i) The ability to correctly integrate proprioception information coming from the periphery might be a key feature for the correct update of the motor control loop, towards the carryover effect, as indicated by the reduced sensitivity to proprioception input to S1 of FES non-responders; (ii) The motor control and FES related rehabilitation induced neural plasticity support the active inference account for motor control, as indicated by the modulation of SMA and M1 connections to S1 area; (iii) SMA has been identified with a dual role of higher order motor processing unit responsible for complex movements, and with a superintendence role in suppressing standard motor plans as external conditions changes. SMA has been shown to be an important asset contributing to the achievement of carry-over effect, following a FES-mediated rehabilitation program. The present study opens the door to predict and functionally explain the achievement of carryover effect following FES-mediated rehabilitation treatment, supporting the clinical need to stratify patients in different rehabilitation programs.

## Figures and Tables

**Figure 1 brainsci-11-00329-f001:**
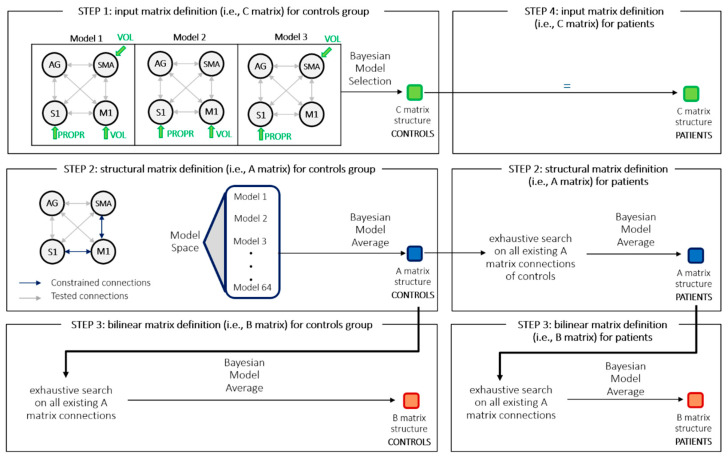
Representation of steps involved in the procedure for model structure identification. STEP 1: input matrix definition (i.e., C matrix) for controls group; STEP 2: structural matrix definition (i.e., A matrix) for controls group; STEP 3: bilinear matrix definition (i.e., B matrix) for controls group; STEP 4: input matrix definition (i.e., C matrix) for patients; STEP 5: structural matrix definition (i.e., A matrix) for patients’ group; STEP 6: bilinear matrix definition (i.e., B matrix) for patients’ group. M1: primary motor cortex; S1: primary somatosensory cortex; SMA: supplementary motor area; AG: angular gyrus; BMA: Bayesian model average.

**Figure 2 brainsci-11-00329-f002:**
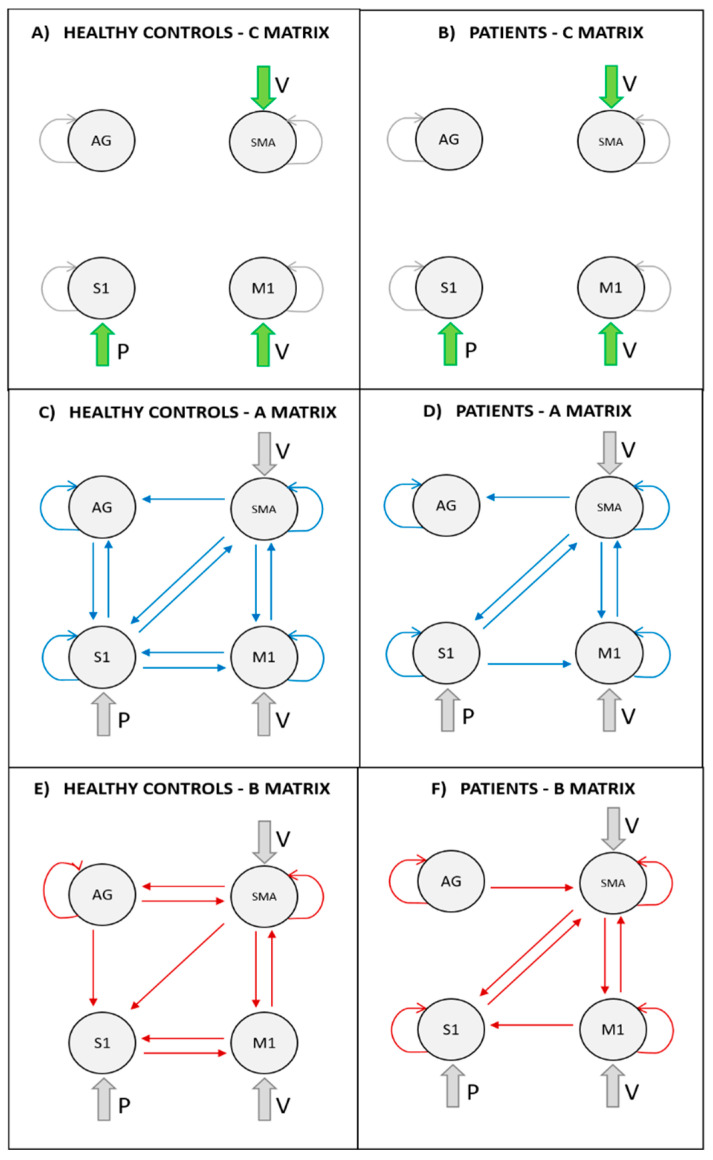
The architecture of the outcome winning model from each methodological STEP. (**A**) STEP 1: winning input matrix (i.e., C matrix) for controls group; (**B**) STEP 4: input matrix for patients group defined as the winning model in STEP 1; (**C**) STEP 2: winning structural matrix (i.e., A matrix) for controls group; (**D**) STEP 5: winning structural matrix for patients group; (**E**) STEP 3: winning bilinear matrix (i.e., B matrix) for controls group; (**F**) STEP 6: winning bilinear matrix for patients group.

**Figure 3 brainsci-11-00329-f003:**
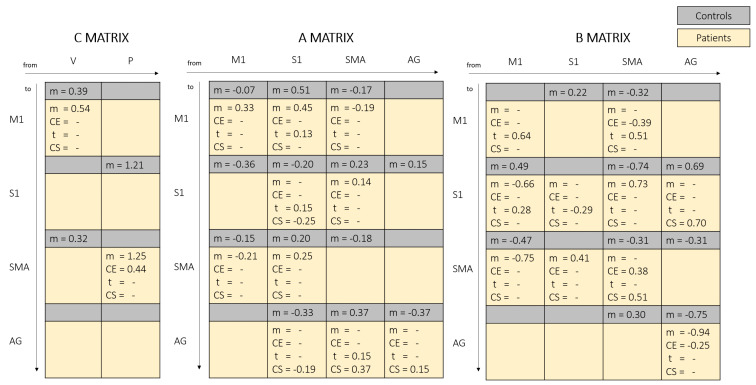
Healthy controls and patients Dynamic Causal Modelling (DCM) estimates in matrix representation. V: voluntary input; P: proprioceptive input; M1: primary motor cortex; S1: primary sensorimotr cortex; SMA: supplementary motor area; AG: angular gyrus; m: mean component; CE: carryover effect regressor; t: time regressor; CS: capacity score regressor.

**Figure 4 brainsci-11-00329-f004:**
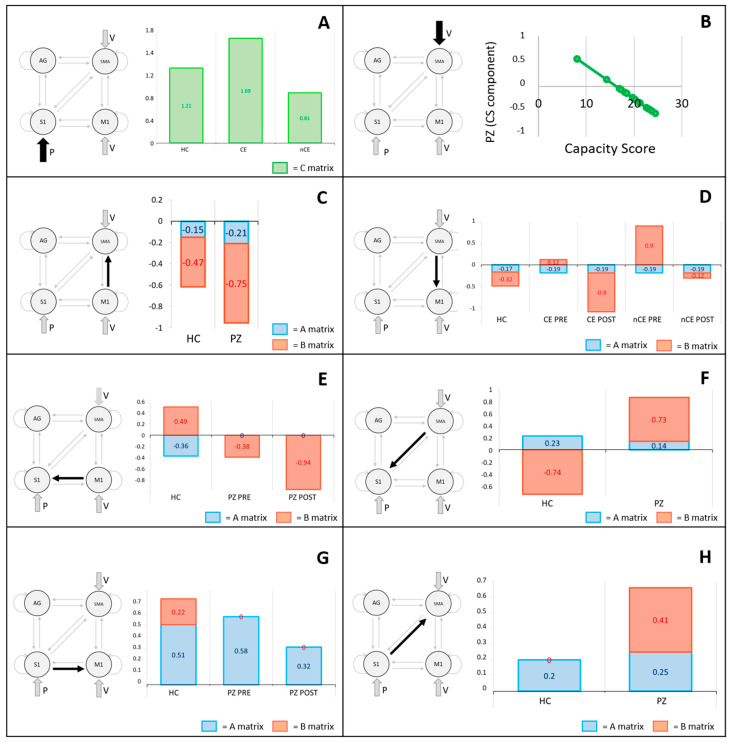
Graphical representation of estimated values for patients and healthy controls. (**A**) input matrix (i.e., C matrix) on S1 area. (**B**) Dependence of the influence of V factor (i.e., C matrix) over SMA area on the capacity score value. Intrinsic and extrinsic estimated parameters (i.e., A matrix – blue bars, B matrix – red bars) for (**C**) M1 to SMA connection; (**D**) SMA to M1 connection; (**E**) M1 to S1 connection; (**F**) SMA to S1 connection; (**G**) S1 to M1 connection; (**H**) S1 to SMA connection. V: voluntary input; P: proprioceptive input; M1: primary motor cortex; S1: primary sensorimotor cortex; SMA: supplementary motor area; AG: angular gyrus; CE: carryover effect responders; nCE: carryover effect non-responders; HC: healthy controls; PZ: patients; “pre” represent to code for pre-training (or baseline) acquired images; “post” represents the code for post-training acquired images.

**Table 1 brainsci-11-00329-t001:** Patients’ individual characteristics. R: right; L: left; MCA: middle cerebral artery; GP: globus pallidus; H: hemorrhagic; I: ischemic.

Subject	Age [Years Range]	Site of Lesion	Type of Stroke	Time [Months]
01	20–25	R MCA	H	23
02	35–40	R GP	I	23
03	60–65	L MCA	H + I	13
04	18–20	L MCA	H	44
05	45–50	L GP	H	44
06	20–25	R MCA	I	30
07	45–50	R GP	I	13
08	60–65	R MCA	H	58

**Table 2 brainsci-11-00329-t002:** Patients’ pre-post clinical scores and carry-over evaluation. TAAI: Tibialis Anterior Activation Index; MRC: Medical Research Council index.

Subj.	Gait Velocity [m/s]	Endurance Velocity [m/s]	Paretic Step Length [mm]	TAAI	MRC	Capacity SCORE	Carry-Over Effect
01	*Pre*: 0.64 *Post*: 0.71	*Pre*: 0.82 *Post:* 0.92	*Pre*: 486 *Post:* 538	*Pre*: 0.60 *Post*: 0.38	*Pre*: 3 *Post*: 4	*Pre*: 20.30 *Post*: 21.32	Yes
02	*Pre*: 0.25 *Post*: 0.27	*Pre*: 0.28 *Post*: 0.24	*Pre*: 194 *Post:* 196	*Pre*: 0.48 *Post*: 0.40	*Pre*: 2 *Post*: 3	*Pre*: 8.15 *Post*: 8.17	No
03	*Pre*: 0.32 *Post*: 0.60	*Pre*: 0.50 *Post*: 0.54	*Pre*: 345 *Post:* 445	*Pre*: 0.56 *Post*: 0.36	*Pre*: 3 *Post*: 3	*Pre*: 14.33 *Post*: 18.15	Yes
04	*Pre*: 0.82 *Post*: 0.86	*Pre*: 1.01 *Post*: 1.10	*Pre*: 561 *Post:* 567	*Pre*: 0.44 *Post*: 0.53	*Pre*: 3 *Post*: 4	*Pre*: 23.42 *Post*: 23.82	No
05	*Pre*: 0.52 *Post*: 0.55	*Pre*: 0.82 *Post*: 0.91	*Pre*: 513 *Post:* 554	*Pre*: 0.46 *Post*: 0.44	*Pre*: 3 *Post*: 4	*Pre*: 21.26 *Post*: 23	Yes
06	*Pre*: 0.70 *Post*: 0.65	*Pre*: 0.98 *Post*: 0.96	*Pre*: 591 *Post:* 544	*Pre*: 0.62 *Post*: 0.71	*Pre*: 3 *Post*: 3	*Pre*: 24.50 *Post*: 22.7	No
07	*Pre*: 0.49 *Post*: 0.48	*Pre*: 0.58 *Post*: 0.63	*Pre*: 420 *Post:* 410	*Pre*: 0.43 *Post*: 0.22	*Pre*: 3 *Post*: 3	*Pre*: 17.31 *Post*: 16.99	No
08	*Pre*: 0.40 *Post*: 0.55	*Pre*: 1.36 *Post*: 1.33	*Pre*: 430 *Post:* 460	*Pre*: 0.69 *Post*: 0.75	*Pre*: 2 *Post*: 3	*Pre*: 18.50 *Post*: 19.76	Yes

**Table 3 brainsci-11-00329-t003:** Mean ROI coordinates of patients and controls ± standard deviation. M1: primary motor cortex; S1: primary somatosensory cortex; SMA: supplementary motor area; AG: angular gyrus.

	PATIENTS			CONTROLS		
	x [mm]	y [mm]	z [mm]	x [mm]	y [mm]	z [mm]
**M1**	−4.3±3.9	−30.8±9.9	−64.8±10.9	−4.3±3.2	−23±10	−67±8.1
**S1**	−7.6±3.8	−43.9±10.6	−66.8±9.2	−4.4±4.3	−35.4±9.3	−67.4±6.5
**SMA**	−4±2.5	−10.1±13.4	−65.8±6.4	−3.3±2.7	−11.5±9.4	−67.4±6.1
**AG**	−37.8±10.1	−67.5±6.4	−43±9.4	−46.9±7.2	−59.1±9	−33.6±12

## Data Availability

The data presented in this study are available on request from the corresponding author. The data are not publicly available due to ethical reasons.

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
