# Peer review of "Brain Plasticity Mechanisms Underlying Motor Control Reorganization: Pilot Longitudinal Study on Post-Stroke Subjects"

_brainsci, 2021, doi:10.3390/brainsci11030329_

Round 1

Reviewer 1 Report

well done, very interesting and useful

Reviewer 2 Report

In this manuscript the authors describe a pilot longitudinal study in which 8 patients after a first-ever stroke, and 16 healthy subjects were investigated. Connectivity network of the carry-over effect after FES applied to the M. tibialis anterior was analysed for responders (n=4) and non-responders (n=4) in patients, and healthy adults. Functional assessment was performed before and after a 4-week treatment phase. ROIs were: M1, S1, SMA and the angular gyrus. These ROI were analysed with dynamic causal modelling analyses.

The manuscript is well written and well structured. It needs only some minor language/punctuation copyediting. The research topic is of high relevance to the field of neurorehabilitation to better understand the effectiveness of therapeutic interventions, specifically to better understand the responsiveness to specific interventions like FES.

However, for my review I will not go into much detail as I believe that the authors should revise their manuscript first to appropriately address their former publication on the same participants and the same topic (M. Gandolla, N. S. Ward, F. Molteni, E. Guanziroli, G. Ferrigno, and A. Pedrocchi, “The Neural Correlates of Long-Term Carryover following Functional Electrical Stimulation for Stroke,” Neural Plast., vol. 2016, p. 4192718, 2016, doi: 538 10.1155/2016/4192718). The authors need to introduce the former results and the benefit of the new analysis regarding their research question. 

Reviewer 3 Report

It would be very helpful to be able to select patients who will benefit from therapy because of the cost and to avoid disappointment. This paper demonstrates a relationship between whether the stroke patient shows a carry-over effect and the connection between the proprioception inputs and S1. If this can be measured in one fMRI session (without any therapy sessions), it may be of great significance in healthcare.

I was disappointed that the site of the neuroplastic change was not discussed. Rushton (your reference 16) hypothesized that the change was occurring in the spinal cord and perhaps your results can refute that idea. What can we learn from your Figure 4D? It looks as if the nCE subjects change their B matrix to near normal (HC) after treatment while the CE group make a change of similar magnitude but going more negative than the HCs. What does this mean?

I found the paper difficult to understand. This is probably partly due to it following several other papers (which I have not read), and partly because I am not familiar with Bayesian methods. However, I think you should try to improve the paper to make it easier to follow. Here are some suggestions.

  1. Too many acronyms. If you must have so many, provide a list. Some are obvious but some are surprising (e.g. PZ = patients). Some seem to be wrong (e.g. FV and FP in legend of Fig 3: should they be EV and EP?).
  2. Section 3.2 is difficult to understand (lines 275-281). What are these (m, CE, t and CS)? Are they numerical variables? Is time the months since stroke? Where are these underscores to be seen?
  3. Table 1 is a pain to look at. Surely you can lay it out better within the constraints of the journal’s rules?
  4. Figure 2 does not let one compare healthy controls against patients easily.  Why not rearrange it  A D; B E; C F instead of A B; C D; E F?
  5. Say somewhere how you can represent each of the matrices by one number.
  6. Explain why, in Figure 4 bar charts, the pink and blue bars are standing one on top of the other instead of side by side.
  7. Line 117. “levels present and absent” would be better as “with stimulator on or off”.
  8. You present the voxel sizes but not the sampling rate.
  9. Line 205 “confound”: what did you mean here?
  10. Line 275: “The coordinates of the ROIs were consistent across subjects (Table 2)” What does this mean? We see the coordinates in the table but what difference would make these inconsistent?
  11. There are lots of spelling mistakes which WORD would show you. (I liked ‘numerosity’ which is a new word for me).
